# The Presence of Hyperreflective Foci Reflects Vascular, Morphologic and Metabolic Alterations in Retinitis Pigmentosa

**DOI:** 10.3390/genes13112034

**Published:** 2022-11-04

**Authors:** Clemens Diem, Cengiz Türksever, Margarita G. Todorova

**Affiliations:** 1Department of Ophthalmology, Cantonal Hospital St. Gallen, 9007 St. Gallen, Switzerland; 2VISTA Clinic Binningen, 4102 Binningen, Switzerland; 3Department of Ophthalmology, University of Zürich, 8006 Zürich, Switzerland; 4Department of Ophthalmology, University of Basel, 4001 Basel, Switzerland

**Keywords:** hyperreflective foci, macular edema, morphologic alterations, metabolic function, retinitis pigmentosa, retinal vessel diameter, retinal vessel oxygen saturation

## Abstract

Background: The presence of hyperreflective foci (HRF) in retinitis pigmentosa (RP) is a potentially new finding. We investigated the presence of HRF in SD-OCT images in eyes with RP and its relation to vascular, morphologic and metabolic findings in RP. Methods: The study was performed on 42 RP patients and 24 controls. Using SD-OCT, we calculated the amount of HRF within the entire retina (HRF-ER) and the outer nuclear layer (HRF-ONL). Retinal vessel diameters (μm) and oxygen saturation (%) values were measured using Oxymap T1. We evaluated the mean diameter in retinal arterioles (D-A) and venules (D-V), the corresponding oxygen saturation values (A-SO_2_, V-SO_2_) and the oxygen saturation difference (A-V SO_2_). Results: RP differed from controls by HRF-ER, HRF-ON and EZ-length (*p* < 0.001). D-A and D-V were narrower and A-SO_2_ and V-SO_2_ were higher in RP (*p* ≤ 0.001). Within RP, significant interactions were found between the HRF-ER* group and: BCVA, EZ length, D-A, A-SO_2_ and A-V SO_2_ (*p* ≤ 0.018). The HRF-ONL* group interactions were significant for: BCVA, EZ length, D-A, A-SO_2_ and A-V SO_2_ (*p* ≤ 0.014). Conclusion: The present study highlights the presence of HRF to reflect the vascular, morphologic and metabolic alterations in RP. These biomarkers seem to be associated with remodeling and apoptosis that occur with the progression of degeneration.

## 1. Introduction

The most apparent and frequently recognized form of inherited retinal disease is retinitis pigmentosa (RP). Its prevalence is accounted to be approximately 0.025% and is nowadays one of the leading causes of blindness within the working population [1,2,3]. RP covers a heterogeneous group of hereditary retinal diseases, which causes at early stages night blindness and progressive visual field constriction. At later stages with impairment of the central retina the visual acuity, color vision and contrast sensitivity drop down [4,5,6,7,8].

Typical signs in patients with RP include bone spicule retinal pigmentary changes progressing from the periphery centripetally, a waxy optic disc and a cystoid macular edema (ME) [9,10,11,12]. The diagnosis of RP is based on clinical features supported by electrodiagnostic findings, such as electroretinogram, fundus photography, fundus autofluorescence imaging, and optical coherence tomography (OCT). All these measurements are defined as “gold standards” for the diagnosis, statement, and evaluation of the disease progression in RP [13,14,15,16,17,18,19,20,21].

A further hallmark is the retinal vessel attenuation, affecting primarily the arteriolar, and later on the venular vessels [22,23,24].

One novel biomarker measuring the metabolic demand in RP is retinal vessel oxygen saturation [25]. Using retinal vessel oximetry (RO), the oxygen saturation of the retinal vessels could be measured in vivo, non-invasively [25,26,27,28,29,30]. Recent RO studies on adults and children affected by RP showed an altered oxygen metabolism, by means of increased retinal vessel oxygen saturation and reduced oxygen consumption [31,32,33,34,35]. As the highest amount of oxygen is used by retinal photoreceptors, a reduction in retinal oxygen demand with a secondary increase in the retinal vessel saturation values following cellular apoptosis has been hypothesized [22,23,33,34,36,37].

Much attention is given nowadays to the presence of intraretinal hyperreflective foci (HRF) in OCT images as an additional well-recognized sign in RP [38,39,40,41]. Their presence has been linked to the extent of intraocular inflammation and the progression of photoreceptor degeneration in RP [39,41].

Since the introduction of retinal gene therapy, it is questionable which patient may benefit and at which stage of degeneration the therapy should be implemented. In addition, apart from dose recommendation, the route of administration, as well as, evaluation of the local and systemic immune responses after ocular gene therapy, are mandatory [42].

Several preclinical studies showed that gene therapy, for example, using rAAV (recombinant adeno-associated virus) might induce retinal toxicity. The latter is confirmed by means of loss of photoreceptors causing the thinning of the outer nuclear layer (ONL) or migration of retinal pigment epithelium (RPE) cells into the neuroretina [42,43]. Thus, currently, reliable morphological signs are necessary for evaluating the effect of therapy but also any possible retinal toxicity in preclinical studies and clinical trials.

Thus, in the present study, we aimed to evaluate to what extent the presence of HRF in RP may reflect the vascular, morphologic and metabolic findings by means of retinal vessel attenuation, the presence of ME and/or epiretinal fibrosis and retinal vessel oxygen saturation.

## 2. Materials and Methods

### 2.1. Subjects

The demographic data of our RP patients and the control group are presented in Table 1. All patients and controls were examined from September 2019 until August 2021 in a single center at the Department of Ophthalmology at Cantonal Hospital St. Gallen, Switzerland. All procedures performed in this study involved human participants and were in accordance with the ethical standards of the institutional/national research committee (EKNZ BASEC 2016-01054) and with the 1964 Helsinki declaration and its later amendments or comparable ethical standards.

Patients and controls underwent all required standard evaluations by an experienced specialist for inherited retinal diseases (M.G.T).

All RP patients presented with a clinical picture of the disease. The clinical phenotype of RP patients was characterized according to the typical funduscopic findings, diagnostic imaging (color fundus photography, fundus autofluorescence and optical coherence tomography; OCT), and electrophysiology (reduced and/or scotopic negative full-field electroretinogram).

All participants underwent a detailed ophthalmic examination, which included, among others, refraction best corrected visual acuity (BCVA, Snellen charts), intraocular pressure examination by applanation, biomicroscopy, and funduscopy. Prior to diagnostic imaging, pupils were maximally dilated (eye drops prepared in our institutional pharmacy as a combination of Phenylephrine HCI 25 mg/mL and Tropicamid 5 mg/mL in 3 mL containers labeled MIX).

Inclusion criteria for controls were Caucasian and best corrected Snellen visual acuity at distance >0.8, refractive error (spherical equivalent) of <6 diopters.

Exclusion criteria for patients and controls were: the above-mentioned inclusion criteria not fulfilled, as well as the presence of ocular and/or systemic pathology other than RP (for instance diabetes mellitus, hypertension, or other metabolic and neurodegenerative diseases, potentially affecting the vessel diameter and oxygen saturation), unstable fixation, and retinal vessel imaging and/or OCT images with inadequate quality.

### 2.2. Retinal Vessel Imaging

Briefly, optic disc-centered fundus images, with a 50° field span, were taken for each eye, starting with the right eye, with the spectrophotometric oximetry unit for retinal vessel oximetry (Oxymap T1). Fundus images were recorded using a camera system, Digital Camera Topcon TRC 50DX. At least two images were taken and more images were added, if necessary. The image with optimal illumination set at least 8% the scale was further selected for analysis. An optic disc-centered image protocol was used where two concentric rings were created in the peripapillary area: one with a radius of 2 optic disc diameters, and the other with a radius of 3 optic disc diameters. The region between these two circles defined the area of interest where all measurements were obtained (Figure 1). The software (Oxymap ehf.) calculates then the optical density ratio (ODR) of the two images (isobestic and non-isobestic) and thus, the mean oxygen saturation of the evaluated retinal vessel. All main arterioles and venules were selected manually within the area of interest. The global mean average arteriolar and venular vessel diameter (D-A and D-V, µm) and the mean arteriolar and venular SO_2_ (A-SO_2_ and V-SO_2_, %) were obtained by simultaneously selecting the main vessels in all four quadrants. Their difference, known to be proportional to the oxygen saturation of the retina (A-V SO_2_, %), was calculated, as well.

### 2.3. Optical Coherence Tomography (OCT)

For the evaluation of the retinal structure, a spectral domain optical coherence tomography (Heidelberg Engineering HRA+OCT Spectralis, Spec-KT-06957, Heidelberg Engineering GmbH, Heidelberg, Germany) was achieved. All images were performed with the macular thickness protocol (OCT Protocol: 1024 × 496 (HR) 20° (6.3 mm); ART (23); Q: 41) and high-definition image protocol (HD 5 Line Raster), which allow high resolution image quality. In both, controls and RP patients the amount of hyperreflective foci along the horizontal whole 3rd single scan were calculated: within the central entire retina (HRF-ER) and the outer nuclear layer (HRF-ONL). In the present study, all HRFs within the central horizontal single HD OCT scan, were counted. The size of the HRF was not taken into account.

### 2.4. OCT Measurements

The presence of HRF was defined based on hyperreflective tiny particles located within the retina. We evaluated their presence and counted them manually within the 2500 [μm] from the fovea and along the horizontal scans. A subgroup analysis was then performed taking the entire retina thickness into account (HRF-ER) and separately along the outer nuclear layer (HRF-ONL; Figure 2). The counts were checked blindly according to the four-eye principle. In addition, horizontal OCT volume scans along the fovea were performed to measure the length and continuity of the ellipsoid zone (EZ), as well as, to examine for the presence of ME and epiretinal fibroplasia (ERF; Figure 2).

### 2.5. Statistical Analysis

For statistical evaluation, ANOVA-based linear mixed-effect models were used with SPSS^®^ (IBM SPSS Statistics^®^, International Business Machines Corp., Armonk, NY, USA; Version 25.0.0.0) which allows for taking the dependency of the left and right eye in the same subject into account and is suitable for repeated measurements, in addition to both eyes being analyzed. A linear mixed-effects model was performed for each pair of the tested methods, where one parameter of the tested pair is a dependent variable.

The mean vessel diameter measurements (D-A and D-V) and retinal vessel oxygen saturation values (A-SO_2_, V-SO_2_ and A-V SO_2_), were taken as independent variables.

In the present study, to predict the effect of RP on HRF count and on retinal vessel diameter and oxygen saturation measurements, the ‘subject’ was taken as a random factor, and the ‘group’, and the ‘eye’ were taken as fixed factors. To exclude the effect on age, the presence of macular edema and the presence of ERF we took them as covariates.

In addition, to predict the effect of HRF count, retinal vessel diameter or oxygen saturation on structural alterations (length and continuity of ellipsoid zone [μm] from the fovea, ME and ERF), the eye and the group-effects were taken into account. In our ANOVA-based mixed-effects model, the eye and the groups were treated as fixed factors and the subject as a random factor. Results are presented as adjusted means and SD for controls with the respective *p*-Values. *p* < 0.05 was defined as statistically significant.

## 3. Results

In total, 132 eyes of 66 subjects were enrolled in the study: 84 eyes of 42 patients diagnosed with retinitis pigmentosa were compared to 48 eyes of 24 age-matched controls. Participants were enrolled based on the criteria described in Section 2. All demographic characteristics of our participants are summarized in Table 1.

In general, the RP group differed from the controls by its structural, functional and metabolic findings (Table 2).

### 3.1. HRF in RP Patients against Controls (OCT Imaging)

The RP group revealed clear differences compared to the controls concerning the presence of HRF-ER and HRF-ONL (*p* < 0.001; Figure 3). The numbers of the HRF-ER in the control group were 36.83 (±14.14), and in patients with RP 100.45 (±49.77). The numbers of the HRF-ONL were 8.72 (±6.97) in the control group and in the RP group 31.4 (±20.68).

In order to link the amount of intraocular affection as part of the photoreceptor degeneration in RP we calculated the HRF-ONL/HRF-ER ratio. Again, RP patients could well be defined from controls by this sign (*p* = 0.001).

### 3.2. Retinal Vessel Diameters in RP Patients against Controls

The mean (±SD) peripapillary retinal vessel diameters (D-A and D-V) were narrower in the RP group than in controls (*p* < 0.001; Figure 4). For instance, in RP the mean (±SD) D-A and D-V were 92.79 μm (±18.50 μm) and 123.31 μm (±22.48 μm) compared to the respective ones in controls 116.77 μm (±9.70 μm) and 158.09 μm (±16.26 μm).

### 3.3. Retinal Vessel Oxygen Saturation Values in RP Patients against Controls

The mean (±SD) A-SO_2_ and V-SO_2_ were higher in patients with RP [101.37% (±5.85%) and 65.73% (±5.87%), respectively] when compared to controls [98.60% (±2.13%) and 60.99% (±3.12%), respectively; the *p*-Values between groups were *p* = 0.049 and *p* = 0.001, linear mixed-effects model]. The corresponding A-V SO_2_ difference, which tended to be lower in RP [35.64% (±7.12%)] compared to controls [37.61% (±2.87%)], did not reach statistically significant values (Figure 5).

### 3.4. Further Structural Alterations Measured by OCT Imaging

#### IS-OS Line/Ellipsoid Zone (EZ)

The EZ line within the 5000 μm length of horizontal OCT scan was intact in the control group, whereas in the RP patients it showed interruptions and was significantly shorter at 2511.28 μm (±1837.86; *p* <0.001).

Furthermore, RP patients differed from controls by the presence of macular edema and/or epiretinal fibrosis. From a total of 84 eyes of 42 patients diagnosed with retinitis pigmentosa 8 eyes of 8 patients had macular edema, 26 eyes of 26 patients had ERF, 5 eyes of 5 patients were part of both ME and ERF RP subgroups. In one patient, both eyes showed the presence of ERF and ME.

### 3.5. Interactions between the Number of HRF and the Functional (BCVA), Structural (Retinal Vessel Diameters and EZ Length) and Metabolic Alterations (Retinal Vessel Oxygen Saturation Values) in RP Compared to Controls

Aiming to predict the effect of the number of HRF on the functional (BCVA), structural (retinal vessel diameter and EZ length) and metabolic (retinal vessel oxygen saturation values) alterations a linear mixed-effects model, was performed.

The interaction effect possibly indicates a different dependence of the number of HRF (HRF-ER and HRF-ONL) on the functional and structural variables between study groups. Therefore, we started with evaluation of the interaction HRF* group effect against the functional (BCVA), structural (retinal vessel diameter, the length of the ellipsoid zone [μm] from the fovea, as well as, the presence of macular edema or ERF) and metabolic findings data (retinal vessel oxygen saturation; Table 3). This means the HRF* group is taken as an independent variable, and the functional, structural and metabolic parameters are taken as dependent variables.

For the average values, the HRF-ER* group interactions were statistically significant only within RP (Table 3) for the following parameters: BCVA (*p* < 0.001), D-A (*p* = 0.018), EZ length (*p* < 0.001) and A-SO_2_ (*p* = 0.002) and A-V SO_2_ (*p* = 0.005).

For the average values, the HRF-ONL* group interactions were statistically significant only within RP (Table 3) for the following parameters BCVA (*p* < 0.011), D-A (*p* = 0.002), EZ length (*p* < 0.002) and A-SO_2_ (*p* = 0.008) and A-V SO_2_ (*p* = 0.014).

In the presence of macular edema in the RP group the higher amount of HRF-ER as well as of HRF-ONL correlated significantly with the BCVA, EZ-length (*p* < 0.001), but also with the A-V SO_2_ (*p* < 0.049). In addition, the higher HRF-ER count correlated with the presence of ME (*p* = 0.013), as well as with the simultaneous presentation of ERF*ME (*p* = 0.017).

For both, RP group and controls significant interactions were found between the number of HRF-ER and HRF-ONL (*p* < 0.001). This means, that when HRF were present in the ONL, they were found in the entire retina.

## 4. Discussion

In agreement with the data to date [38,39,40,41], the current study confirms again, that RP patients have an increased number of HRF compared to controls. Furthermore, the present study shows that the retinal vessel diameters are affected in RP [22,23,24]. Consistent with the results of published studies [22,23,33,34,36,37], we found once again an increased A-SO_2_ and even more, an increased V-SO_2_ in in RP group.

Considering the above-cited parameters, the RP group could be differentiated from controls by these signs. Furthermore, in our study, the presence of HRF correlated well with the degree of retinal vessel attenuation, the presence of ME, epiretinal fibrosis, the degree of vision deterioration and the extends of metabolic alteration. There are several explanations for the above-cited findings.

### 4.1. HRF in RP Patients

HRFs are well-circumscribed lesions located in the retina or choroidea, which are observed in a variety of retinal diseases. In the SD-OCT scan they are seen as tiny dense particles in the outer- and inner retinal layers with greater reflectivity than the retinal pigment epithelium (RPE). Their prevalence has been related to the progression of atrophy in age-related macular degeneration [44], to the activity of diabetic retinopathy [45,46,47], macular telangiectasia type 2 (MacTel) [48] or Morbus Coats [49], but also to the extent of ischemia in retinal vein occlusion [50], to the progression of degeneration in RP [38,39,40,41], as well as to the progression of atrophy in Stargardt disease [51,52].

Regarding RP patients, several studies have reported the HRFs to be available not only in the outer retinal layer but also in the choroid [38,39,40,52]. The present study confirms again, that RP patients have an increased number of HRFs compared to controls. This was the cause when evaluating the entire thickness of the retina (HRF-ER), as well as limiting only to the outer nuclear layer (HRF-ONL): The RP group revealed clear differences compared to the controls. Thus, the increased number of HRF seems to be a valuable sign of retinal degeneration in RP patients.

There are several hypotheses explaining the origin of HRF in general. Some studies suppose that the HRF represents an inflammatory component like microglia or migrated RPE and/or the extravasation of lipoproteins following the breakdown of the blood-retinal barrier [53,54]. It has been shown that HRF foci are also a predictive marker of the final visual outcome in diseases like retinal vein occlusion [50], diabetic retinopathy [46], polypoidal choroidal vasculopathy [55], neovascular age-related macular degeneration [55], but also in degenerative diseases like RP [38,39,40,41].

In retinitis pigmentosa, based on the results of immunohistochemistry, other studies suppose the presence of HRF to reflect the degenerative process in response to apoptosis. In cadaver eyes with RP, a prominent neurite spouting of the rod, amacrine and horizontal cells, have been confirmed. These changes in the inner retina have been discussed to be a manifestation of regeneration on the part of the remaining inner retina even at the expense of normal visual function [56].

Following that, and supported by other histopathology studies [57,58], the pattern of the HRF distribution in RP has been hypothesized to be associated with the progression of degeneration [38]. The presence of HRF only in the inner nuclear layer was supposed to represent the neurovascular remodeling in the early stage of the degeneration. With its progression and migration of RPE cells into the outer retina, HRF has been detected in the outer nuclear layer [38,39]. According to the above-cited study and based on histopathological findings, at a far late stage of degeneration and further thinning of the inner retina HRF [57,58], seems to disappear.

Noticeably, the number of HRF in the outer nuclear layer [41] has been linked to the amount of intraocular inflammation and photoreceptor degeneration in RP by means of aqueous flare value, EZ disruption and areas of low autofluorescence. Therefore, the authors concluded the presence of HRF in the outer nuclear layer reflects the RPE cell—or microglial migration in response to photoreceptor degeneration [41].

In agreement, we also found a correlation between the HRF-ER* group and the EZ length (*p* < 0.001). The same held true for the HRF-ONL* group interactions with the EZ length (*p* < 0.002). Correspondingly, we revealed significant interactions between the HRF-ER* group and BCVA (*p* < 0.011), but also HRF-ONL* group and BCVA (*p* = 0.011).

Thus, we suppose, that the presence of HRF in RP corresponds to the EZ disruption as well as to the BCVA reduction, as part of the degenerative process.

Furthermore, in our study, we found that the higher HRF-ER count in RP was linked to the presence of ME (*p* = 0.013) but also to the simultaneous presentation of ERF*ME (*p* = 0.017). These findings therefore suppose the presence of ME and/or ERF to be a part of remodeling in response to more advanced stage of photoreceptor degeneration.

### 4.2. Attenuated Retinal Vessels in RP Patients

A further finding of our study is the attenuated peripapillary retinal vessels in RP eyes when compared to the controls. Our results are in accordance with several studies, which also found the diameter of the retinal vessels to be narrower in RP in comparison to controls [22,23,24]. The authors explained these findings with the assumption for complex vascular impairment is thought to be induced by neurovascular remodeling. According to this data, the amount of retinal vessel attenuation should be proportional to the tissue remodeling occurring in RP patients [59,60,61] and thus, to the residual retinal function and structure [22,23,24,62]. Our RP group was distinguished from controls through this sign also.

A novel finding in the current study is the link found between retinal vessel attenuation and the presence of HRF in RP patients. This association has not been investigated, so far. We found within our RP patients that the HRF-ER* group interactions were statistically significant against the D-A (*p* = 0.018; Table 3). Furthermore, the HRF-ONL* group interactions were statistically significant for D-A (*p* = 0.002; Table 3). Altogether, our results indicate a clear relationship of the presence of HRF and the examined retinal vessel attenuation parameters, for the RP versus controls.

### 4.3. Metabolic Alterations in RP Patients

Consistent with the results of previous studies [22,23,33,34,36,37] our RP patients showed again altered metabolic function by means of increased oxygen saturation of retinal arterioles and venules. Oxygenation of the human retina is a dynamic process, and regulation is necessary to provide a healthy metabolic environment [63]. In the presence of other photoreceptors degeneration and consequently reduced oxygen use, since the oxygen delivery from the outer retina remains unchanged, the intraretinal oxygen levels are expected to increase [64,65,66,67]. Thus, increased retinal vessel oxygen saturation values would be measured in the inner retina, as it is the cause in our study using the Oxymap device.

One novel finding in the current study is the association found between the amount of the HRF-ER as well as the HRF-ONL and the A-SO_2_ as well as the A-V SO_2_ parameters. This means that increased HRF-ER and HRF-ONL reflect increased A-SO_2_ and decreased A-V SO_2_ values or vice versa. One possible explanation of these findings could be the following: with the progression of degeneration and continuous loss of photoreceptors, the increased oxygen level in the retina may reflect the RPE cell—or microglial migration in response to photoreceptor degeneration [41], as already proposed. In addition, increased superoxide radicals lead to the generation of other reactive oxygen species in the retina [68].

The above-cited results are pointing that the presence of increased HRF count, retinal vessel attenuation and increased retinal vessel oxygen saturation values correspond to the progression of the degeneration.

Our entire discussion is in accordance with previous histopathology studies on RP. These studies have hypothesized the degeneration of the inner retinal cells to follow that of the photoreceptors, and to be presumably connected to either transneuronal damage or vascular compromise [69]. Changes in the number of the affected axons, and ganglion cells, but also the narrowing of the retinal vessels crossing the margin of the optic disc are found to contribute to the optic disc pallor in RP patients.

In a patient with RP accompanied with diabetic retinopathy and reduced visual acuity HRF have been described to be present in the retinal and choroidal layers. These data supposed a relationship of their distribution in the choroid to reflect the extent of the disappearance of ELM and EZ line with thinning of the ONL [40]. In the presented case, a kind of choroidal involvement has been hypothesized.

In general, all diseases of the pachychoroidal spectrum have a thickened choroid with attenuation of the choriocapillaris in common without a known common cause. In RP the presence of pachychoroidal disease have revealed several patterns [39]. Some authors have found a reduced lower mean choroidal thickness in patients with RP [52,70], whereas, others to the contrary, an increase in the choroidal thickness [71]. Apart from a normal appearing choroid in RP, choroid with reduced vasculature, as well as, choroid with caverns have been reported. Noticeably, the last type has shown a correlation with a worse visual outcome and decreased perfusion density [72]. Furthermore, in the presence of HRF a decrease in the choroidal vascular index in RP has been found [52].

Therefore, a relationship between RPE, photoreceptors, the choroidal morphology, and choroidal vasculature seems to be related to visual impairment and residual retinal function [73]. Since RP patients were thought to develop a pachychoroid disease with progression of RP degeneration, the presence of HRF in the choroid was supposed to be found in the advanced stages of the degeneration. Nevertheless, Kuroda et al. found that HRFs in the outer retinal layer were more common compared to those in the pachychoroidea, even in the advanced stages of RP [38]. Similarly, we also found in our RP subjects the highest amount of HRFs in the outer retina compared to those in the entire retina. This finding showed again statistically significant difference compared to controls (*p* = 0.001).

Based on the link found in the present study between the amount of HRF, retinal vessel attenuation and reduced oxygen use, the degeneration of photoreceptors seems to lead to structural and vascular remodeling. The measured metabolic demand in patients with retinitis pigmentosa probably serves as an additional hallmark allowing to evaluate the progression of the degeneration. This finding seems to reflect the decreased metabolic demand of the degenerating retina with the secondary remodeling of the inner retina and the choriocapillaris [74,75,76,77]. Previous agreeing studies found a further diminished blood flow in the retinal and choroidal vessels of RP [61,78]. Furthermore, significant narrowing of the peripapillary vessels as well as reduced superior foveolar avascular zone and deep foveolar avascular zone in RP patients were confirmed in a study of Della Volpe et al. [24]. All of these findings supposed reduced metabolic demand as a result of photoreceptors loss in RP.

Several studies have shown that in progressive RP retinal damage morphologic and mictovascular changes are to be expected and found in the macular area. According to these studies the presence of an intact EZ line, the absence of ME and ERF on OCT translated into better BCVA [10,16,21,79,80,81,82]. In agreement with the stated, we found in RP in the presence of macular edema that more attenuated D-A correlated significantly with the amount of HRF-ER (*p* = 0.023). Furthermore, more attenuated D-V correlated well with the amount of HRF-ER and the HRF-ONL (*p* = 0.007 and *p* = 0.008, respectively). All these findings reflect more altered vascular and structural remodeling with progression of photoreceptor degeneration.

## 5. Conclusions

In conclusion, from the present study, we derive clear evidence that an increased number of HRF, attenuated peripapillary vessels, the presence of ME and ERF, and altered metabolic function in RP patients are linked to the more progressive and active stage of degeneration.

This study concludes that by evaluating these parameters in RP a predictive model might be proposed. It would identify which individuals are at risk of developing a more aggressive photoreceptor degeneration of RP, and would suppose which ones may profit from future therapy. With arguably greater sensitivity, it would also point out those RP patients that may possibly develop therapy-induced retinal toxicity.

### Limitations

There are some limitations of the present study, as for instance:Relatively small number of cases due to the rarity of the disease were included in the study.One further limitation is the absence of follow up, which could have helped to reevaluate the progression taking into account the number of hyperreflective foci and visual acuity.Compared to the study of Chu-Hsuan Huang [39] we divided the groups into hyperreflective foci within the entire retina (HRF-ER) and the outer nuclear layer (HRF-ONL) without considering the foci in the choroid and their thickness. The study may also be limited due to the results of Hanumunthadu et al. [52] describing a decrease in the choroidal vascular index in RP patients when HRF were represented.Evaluation of the presence of HRF has been proposed in several studies to be done using the EDTRS grid of the macula. However, we decided to keep the data analysis using horizontal scans, so that to be able to take the presence of ME und ERF in RP into account.One remaining open question in the present study is whether the HRF is also present in healthy controls. The presence of HRF in normal eyes and with aging has not been discussed until now. Some possible explanations could be the decreased vessel dispensability, the ongoing vascular structural remodeling and also changes in viscoelastic properties occurring with ageing [83]. All of these are probably leading to metabolic changes throughout their lifetime [84].

Therefore, further prospective, randomized and controlled studies with a larger number of patients are required.

## Figures and Tables

**Figure 1 genes-13-02034-f001:**
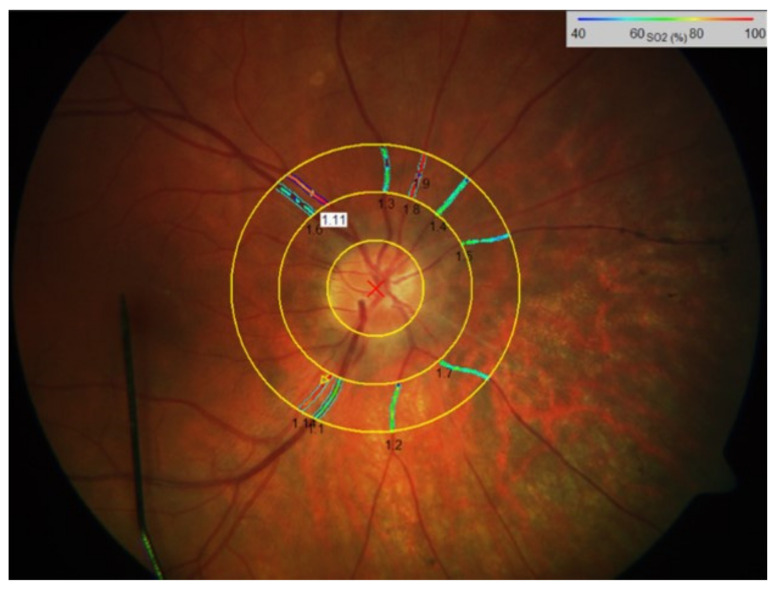
A peripapillary vessel map of the retinal vessel image of a patient with retinitis pigmentosa. Retinal vessel diameters in retinal arterioles (D-A, µm) and retinal venules (D-V, µm), were measured within the peripapillary annulus in control subjects as well as in RP patients. The corresponding retinal vessel oxygen saturation values (A-SO_2_, V-SO_2_ and A-V SO_2_) were calculated, as well. The colors in the oximetry image indicate the relative oxygen saturation (%) in retinal vessels as quantified in the scale bar. Note the increased A-SO_2_ compared to V-SO_2_.

**Figure 2 genes-13-02034-f002:**
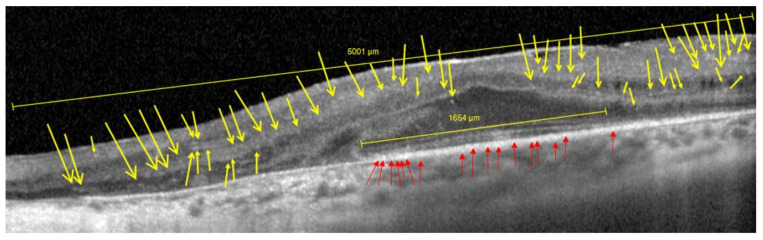
An example of optical coherence tomography (OCT) measurement, using the macular thickness image-protocol (right eye), of an RP patient. Except for the average number of HRF-ER (yellow-colored arrows), we calculated those of the HRF-ONL (red-colored arrows), as well as the length of the EZ (horizontal line, yellow-colored, on the bottom).

**Figure 3 genes-13-02034-f003:**
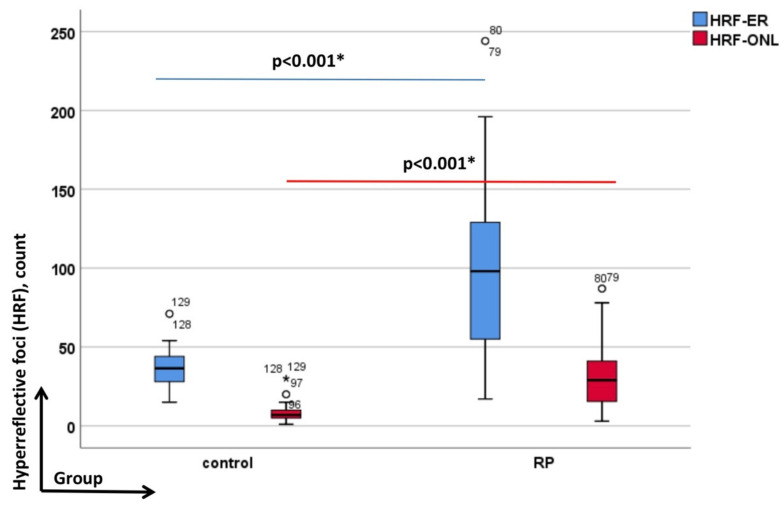
Depicts the data of HTF-ER blue-colored; and of the HRF-ONL red-colored. The box plot is the interquartile range; the short horizontal bold line depicts the median. In each graph the groups as labelled on the y-axis (from left to the right: controls and retinitis pigmentosa (RP) and the evaluated HRF-parameters—on the x-axis. All significant *p*-values are given in bold with asterisk and the numbered circles the outliers.

**Figure 4 genes-13-02034-f004:**
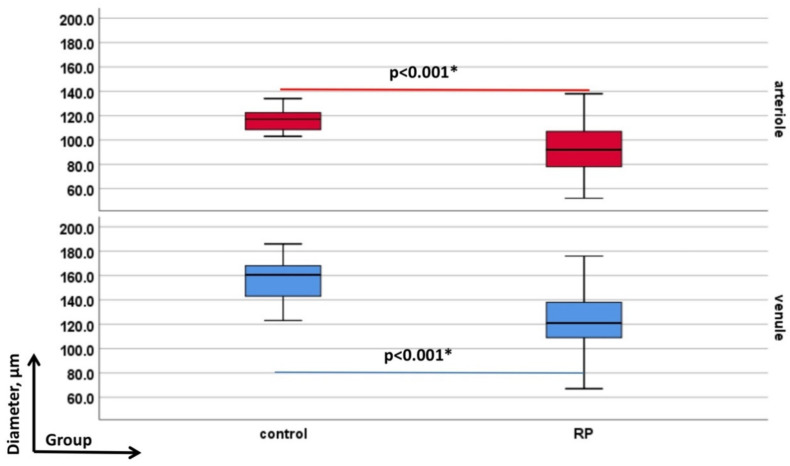
Depicts retinal vessel diameters’ data of arterioles (D-A, red-colored) and venules (D-V, blue-colored). In each graph the groups as labelled on the y-axis (from left to the right: controls and retinitis pigmentosa (RP)) and the evaluated diameter-parameters—on the x-axis. All significant *p*-values are given in bold with asterisk.

**Figure 5 genes-13-02034-f005:**
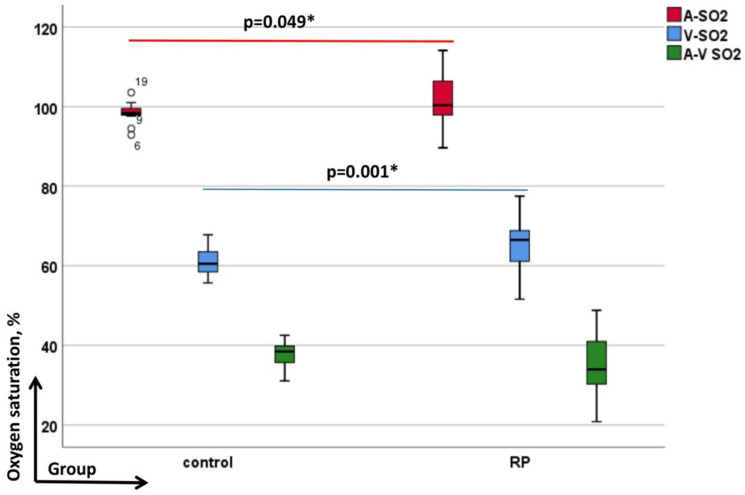
Depicts retinal vessel oxygen saturation data of arterioles (A-SO_2_, red-colored) venules (V-SO_2_, blue-colored) and their corresponding difference (A-V SO_2_, green-colored). In each graph the groups as labelled on the y-axis (from left to the right: controls and retinitis pigmentosa (RP)) and the evaluated diameter-parameters—on the x-axis. All significant *p*-values are given in bold with asterisk and the numbered circles the outliers.

**Table 1 genes-13-02034-t001:** Demographic data of our RP patients and controls.

Demographic Characteristics	Retinitis Pigmentosa Patients	Controls
Number of Subjects (eyes)	42 (84)	24 (48)
Mean Age [± SD]	49.76 [14.94]	42.88 [16.32]
Sex: ♀/♂	20/22	14/8

Abbreviations:♀: female; ♂: male.

**Table 2 genes-13-02034-t002:** depicts the BCVA (Snellen chart), the number of HRF-ER and HRF-ONL, the length of the EZ [µm], the retinal vessel diameters’ (D-A, D-V, [µm]) and retinal vessel oxygen saturations’ (A-SO_2_, V-SO_2_ and A-V SO_2_ [%]) adjusted means with their corresponding standard deviation calculated for the groups, as the average values. The *p*-Values between groups are given in the right columns (ANOVA, based on linear-mixed effects model). Statistical significance is defined as *p* < 0.05.

Parameters	Groups	Mean	±SD	*p*-Values between Controls and RP Patients
BCVA (Snellen chart)	Controls RP	0.92 0.69	0.23 0.36	<0.001
HRF-ER, number	Controls RP	36.83 100.45	14.14 49.77	<0.001
HRF-ONL, number	Controls RP	8.72 31.4	6.97 20.68	<0.001
EZ length, µm	Controls RP	5000 2511.28	0 1837.86	<0.001
D-A, µm	Controls RP	116.77 92.79	9.70 18.5	<0.001
D-V, µm	Controls RP	158.09 123.31	16.26 22.48	<0.001
A-SO_2_, %	Controls RP	98.60 101.37	2.13 5.85	0.049
V-SO_2_,%	Controls RP	60.99 65.73	3.12 5.87	0.001
A-V SO_2_, %	Controls RP	37.61 35.64	2.87 7.12	0.198

Abbreviations: BCVA: best corrected visual acuity; HRF-ER: hyperreflective foci within the entire retina; HRF-ONL: hyperreflective foci within the outer nuclear layer; EZ: ellipsoid zone; D-A: diameter of retinal arterioles; D-V: diameter of retinal venules; A-SO_2_: oxygen saturation of retinal arterioles; V-SO_2_: oxygen saturation of retinal venules; A-V SO_2_: oxygen saturation difference.

**Table 3 genes-13-02034-t003:** presenting interactions between the number of HRF (HRF-ER and HRF-ONL) and the functional, structural and metabolic findings taking the group-effect into account. Only statistically significant values (*p* < 0.05) are given bellow.

Variables	Predictors					
	HRF-ER			HRF-ONL		
	Both RP and Controls	RP	Controls	Both Groups	RP	Controls
IS/OS length [μm] RP-ME RP-no-ME	<0.001	<0.001 <0.001 <0.001		<0.001	0.002 <0.001 0.005	
BCVA [Snellen chart] RP-ME RP-no-ME	<0.001	0.001 <0.001		0.044	0.011 <0.001	
HRF-ONL [count] RP-ME RP-no-ME	<0.001	<0.001 <0.001	<0.001			
HRF-ER [count] RP-ME RP-no-ME				<0.001 <0.001	<0.001	<0.001
D-A [μm] RP-ME RP-no-ME		0.018			0.002 0.008	
D-V [μm] RP-ME RP-no-ME						
A-SO_2_ [%] RP-ME RP-no-ME	0.002	0.006		0.018	0.037	
V-SO_2_ [%] RP-ME RP-no-ME						
A-V SO_2_ [%] RP-ME RP-no-ME	0.005	0.022 0.022		0.014	0.048 0.048	

## Data Availability

Not applicable.

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
