# Peer review of "The Presence of Hyperreflective Foci Reflects Vascular, Morphologic and Metabolic Alterations in Retinitis Pigmentosa"

_genes, 2022, doi:10.3390/genes13112034_

Round 1

Reviewer 1 Report

The authors evaluetas HRFs in retinitis pigmentosa. This is a well performed study, indicating that an increased number of HRFs is indicative of the severity of disease, as in several other retinal disorders.

One thing that is puzzling in this and other studies, is that HRFs are seen in normal probands as well. If HRFs are an indicator of inflammation or degeneration, would this indicate that these processes occur in all normal eyes. Or do the HRFs in normal eyes have another origin compared to eyes with retinal disease? It would be helpful if the authors ould comment on that.

There are a few spelling errors that should be re-checked.

The position of images and legends could be improved through final editing

Author Response

The authors evaluetas HRFs in retinitis pigmentosa. This is a well performed study, indicating that an increased number of HRFs is indicative of the severity of disease, as in several other retinal disorders.

One thing that is puzzling in this and other studies, is that HRFs are seen in normal probands as well. If HRFs are an indicator of inflammation or degeneration, would this indicate that these processes occur in all normal eyes. Or do the HRFs in normal eyes have another origin compared to eyes with retinal disease? It would be helpful if the authors ould comment on that.

  • The presence of HRF in normal eyes and with aging has not been discussed until now. Some possible explanations could be: the decreased vessel dispensability, the ongoing vascular structural remodeling, changes in viscoelastic properties occurring with ageing (Seshadri et al. 2016), all of which probably leading to metabolic changes throughout lifetime (Waizel et al. 2018).
  • Seshadri S, Ekart A & Gherghel D (2016): Ageing effect on flicker-induced diameter changes in retinal microvessels of healthy individuals. Acta Ophthalmol 94: 35-42.
  • Waizel M, Türksever C, Todorova MG. Normative values of retinal vessel oximetry in healthy children against adults. Acta Ophthalmol. 2018 Nov;96(7):e828-e834. doi: 10.1111/aos.13726. Epub 2018 Sep 5. PMID: 30187646.

There are a few spelling errors that should be re-checked.        

  • In agreement with the reviewer, we made a spelling check-up through the manuscript and implemented changes in the revised manuscript.

The position of images and legends could be improved through final editing

  • The format is followed according to the rules of Genes Journal.

Reviewer 2 Report

Authors have provided interesting and stimulating data about hyperreflective foci (HRF) in RP with a potential use such as biomarker in OCT and in other retinal pathologies. However, overall some questions remain on the experimental parts and we have noted the following deficiencies in the manuscript:

Abstract

-Check the number in parenthesis (1), (2), are they necessary in the abstract?

-Check the abbreviations.

-Check the words in the abstract.  The abstract should be a total of about 200 words maximum.

Introduction

-Line 73. It would be interesting to add that the HRF are also present in other retinal diseases and add the corresponding references.

-Line 124.

Materials and Methods

-Line 100. The authors should provide information or mention about the written consent or ethical protocol of the patients.

-Line 124. Retinal vessels imaging.

This aspect is not clear. Are two images used in all cases? In which cases may more than one image be necessary? In the case of 2 images, what statistics were used?

-Line 127-128. Please, explain the thickness of the region used in the study?

It is important to know this thickness and the number of planes used to be able to extrapolate the results. Is this studied area enough to extrapolate the results?

Were the measurements of the all parameters quantified by two investigators who blinded?

A general question is about the size of the HRD measured. What size are the dots? Have they all been included?

Results

-The numbers of patients in the manuscript does not match in table, please, check it.

-Please, delete the title “This is a figure 1”, “this is a figure 2” etc. from figure legends.

-Figure 1. Please, explain the oxygenation level ranges from blue to dark red in the figure legend. Add abbreviations and scale bar.

-Figure 3. Please, add the asterisk significance.

Author Response

Authors have provided interesting and stimulating data about hyperreflective foci (HRF) in RP with a potential use such as biomarker in OCT and in other retinal pathologies. However, overall some questions remain on the experimental parts and we have noted the following deficiencies in the manuscript:

Abstract

Check the number in parenthesis (1), (2), are they necessary in the abstract?

- We excluded in agreement the parenthesis.

Check the abbreviations.

Check the words in the abstract.  The abstract should be a total of about 200 words maximum. 

- We reduced the word count in our abstract to 200:

  • Abstract: Background: The presence of hyperreflective foci (HRF) in retinitis pigmentosa (RP) is a potentially new finding. We investigated the presence of HRF in SD-OCT images in eyes with RP and its relation to vascular, morphologic and metabolic findings in RP. Methods: The study was performed on 42 RP patients and 24 controls. Using SD-OCT, we calculated the amount of HRF within the entire retina (HRF-ER) and the outer nuclear layer (HRF-ONL). Retinal vessel diameters (μm) and oxygen saturation (%) values were measured using Oxymap T1. We evaluated the mean diameter in retinal arterioles (D-A) and venules (D-V), the corresponding oxygen saturation values (A-SO2, V-SO2) and the oxygen saturation difference (A-V SO2). Results: RP differed from controls by HRF-ER, HRF-ON and EZ-length (p<0.001). D-A and D-V were narrower and A-SO2 and V-SO2 were higher in RP (p£001). Within RP, significant interactions were found between the HRF-ER* group and: BCVA, EZ-length, D-A, A-SO2 and A-V SO2 (p£0.018). The HRF-ONL* group interactions were significant for: BCVA, EZ-length, D-A, A-SO2 and A-V SO2 (p£0.014). Conclusion: Our study highlights the presence of HRF to reflect the vascular, morphologic and metabolic alterations in RP. These biomarkers seem to be associated with remodeling and apoptosis that occur with progression of degeneration.

Introduction

Line 73. It would be interesting to add that the HRF are also present in other retinal diseases and add the corresponding references.

- We completely agree with the reviewer’s suggestion. However, in order to reduce the repetition in text, this information is available in the discussion section. “HRFs are well-circumscribed lesions located in the retina or choroidea, which are observed in a variety of retinal diseases. In the SD-OCT scan they are seen as tiny dense particles in the outer- and inner retinal layers with greater reflectivity than the retinal pigment epithelium (RPE). Their prevalence has been related to the progression of atrophy in age-related macular degeneration [44], to the activity of diabetic retinopathy [45-47], macular telangiectasia type 2 (MacTel) [48] or Morbus Coats [49], but also to the extent of ischemia in retinal vein occlusion [50], to the progression of degeneration in RP [38-41], as well as to the progression of atrophy in Stargardt disease [51, 52].”

Line 124. 

Materials and Methods 

Line 100. The authors should provide information or mention about the written consent or ethical protocol of the patients. 

            - Done in agreement.

Line 124. Retinal vessels imaging. 

This aspect is not clear. Are two images used in all cases? In which cases may more than one image be necessary? In the case of 2 images, what statistics were used?

- We clarified this information for the reads in the text: “at least 8% the scale was further selected for analysis… The software (Oxymap ehf.) calculates then the optical density ratio (ODR) of the two images and thus, the mean oxygen saturation of the evaluated retinal vessel.”

Line 127-128. Please, explain the thickness of the region used in the study? 

  • The mean thickness of the macula in patients with retinitis pigmentosa due to atrophy, remodeling or macular edema is either reduced or thickened. Therefore, in this study, we did not measure the mean thickness of the macula. This is a topic of one other study we are performing, where the RP group will be subdivided according to this criterion.
  • This information is included in the methods section.

It is important to know this thickness and the number of planes used to be able to extrapolate the results. Is this studied area enough to extrapolate the results? 

  • We applied HD 5 Line Raster horizontal scans, which allow high resolution image quality. After the image acquisition, the number of HRF were counted by each investigator only on the whole 3th single scan, which is the central one.
  • This information is included in the methods section.

Were the measurements of the all parameters quantified by two investigators who blinded? 

  • This information is added now in the text: “The counts were checked blindly according to the four-eye principle. “

A general question is about the size of the HRD measured. What size are the dots? Have they all been included?

  • In the present study, all HRFs within the central horizontal single HD OCT scan, were counted. The size of the HRF was not taken into account. 
  • This information is included in the methods section.

Results

The numbers of patients in the manuscript does not match in table, please, check it.

            - Thanks reviewer’s remark we corrected the number of subjects and eyes.

Please, delete the title “This is a figure 1”, “this is a figure 2” etc. from figure legends.

- Done in agreement.

Figure 1. Please, explain the oxygenation level ranges from blue to dark red in the figure legend. Add abbreviations and scale bar.

This information is included in the text: The colors in the oximetry image indicate the relative oxygen saturation (%) in retinal vessels as quantified in the scale bar. Note the increased A-SO2 compared to V-SO2.

Figure 3. Please, add the asterisk significance. 

  • Done in agreement

We thank reviewer for her/his helpful suggestions.